ecology/chemical ecology

*Brassica rapa*, *Cotesia vestalis*, *Plutella xylostella*, conservation biological control, herbivory-induced plant volatile, greenhouse

# Targeting diamondback moths in greenhouses by attracting specific native parasitoids with herbivory-induced plant volatiles

Masayoshi Uefune[1,†], Junichiro Abe[2,†], Kaori Shiojiri[3], Satoru Urano[4], Koukichi Nagasaka[5] and Junji Takabayashi[6]

[1]Department of Agrobiological Resources, Faculty of Agriculture, Meijo University, Nagoya, Aichi 468-8502, Japan
[2]Western Region Agricultural Research Center, National Agriculture and Food Research Organization, Fukuyama, Hiroshima 721-8514, Japan
[3]Department of Agriculture, Ryukoku University, Otsu, Shiga 520-2194, Japan
[4]Peco IPM Pilot Co. Ltd., Kumamoto, Kumamoto 860-0004, Japan
[5]Central Region Agricultural Research Center, National Agriculture and Food Research Organization, Tsukuba, Ibaraki 305-8666, Japan
[6]Center for Ecological Research, Kyoto University, Otsu, Shiga 520-2113, Japan

MU, 0000-0002-9570-8938; KS, 0000-0003-4972-6454;
KN, 0000-0001-8780-3962; JT, 0000-0002-3575-9952

**Author for correspondence:**
Junji Takabayashi
e-mail: takabayashi.junji.4a@kyoto-u.ac.jp

[†]Equal contributions.

We investigated the recruitment of specific parasitoids using a specific blend of synthetic herbivory-induced plant volatiles (HIPVs) as a novel method of pest control in greenhouses. In the Miyama rural area in Kyoto, Japan, diamondback moth (DBM) (*Plutella xylostella*, Lepidoptera: Plutellidae) larvae are an important pest of cruciferous crops in greenhouses, and *Cotesia vestalis* (Hymenoptera: Braconidae), a larval parasitoid of DBM, is found in the surrounding areas. Dispensers of HIPVs that attracted *C. vestalis* and honey feeders were set inside greenhouses (treated greenhouses). The monthly incidence of DBMs in the treated greenhouses was significantly lower than that in the untreated greenhouses over a 2-year period. The monthly incidences of *C. vestalis* and DBMs were not significantly different in the untreated greenhouses, whereas monthly *C. vestalis* incidence was significantly higher than monthly DBM incidence in the treated greenhouses. Poisson regression analyses showed that, in both years, a significantly higher number of *C. vestalis* was recorded in the treated greenhouses than in the untreated greenhouses when

the number of DBM adults increased. We concluded that DBMs were suppressed more effectively by *C. vestalis* in the treated greenhouses than in the untreated greenhouses.

# 1. Introduction

In response to infestation by herbivorous arthropods, plants emit so-called herbivory-induced plant volatiles (HIPVs) that attract natural enemies (i.e. predators and parasitoids) of the herbivores [1–5]. The emission of HIPVs may be an induced indirect defence strategy of plants against herbivores. Attracting higher densities of natural enemies into agricultural areas using this induced indirect defence strategy might be one way to effectively control pests. Previous applied studies on the use of this defence strategy by plants include the application of HIPVs commonly released by a wide range of plants infested by herbivorous arthropods, and the use of plant hormone analogues that induce plants to emit HIPVs (e.g. methyl jasmonate and methyl salicylate) for the attraction of unspecified natural enemies (for reviews, see [4,6–8]).

An intriguing observation about HIPVs is that plants can attract the specific natural carnivorous enemies of currently infesting herbivorous pests by emitting herbivore species-specific blends of HIPVs [1–5]. However, to date, there have been no attempts to use this specificity for pest control, i.e. to control target pest insects in a particular location by attracting specific native natural enemies using species-specific blends of HIPVs. The objective of this study is to test this possibility.

The larvae of the diamondback moth (DBM) (*Plutella xylostella*) are important pests of cruciferous crops worldwide [9]. *Cotesia vestalis* is a solitary parasitoid wasp that can be effective in the control of DBM larvae [9–12]. The specific response of *C. vestalis* to DBM larvae-HIPVs under both laboratory and field conditions has been reported in previous studies [13–17]. *Cotesia vestalis* females show a specific olfactory response toward DBM-infested cabbage plants [13]. A synthetic blend of four HIPVs [(Z)-3-hexenyl acetate, α-pinene, sabinene and *n*-heptanal] emitted from DBM-infested cabbage plants has been shown to attract *C. vestalis* females under laboratory, greenhouse and experimental field conditions [14–17].

We conducted field studies in the Miyama rural area in Kyoto Prefecture, where DBM larvae are important pests of cruciferous crops in greenhouses, and populations of DBM and *C. vestalis* are present in the surrounding areas [18]. The objective of this study was to test whether a conservation biological control strategy combining the artificial attraction of native *C. vestalis* from surrounding areas using a synthetic blend of HIPVs with the subsequent artificial feeding of *C. vestalis* could be an effective method for the control of DBM populations in cruciferous crop greenhouses.

# 2. Materials and methods

## 2.1. Herbivory-induced plant volatile dispensers

Four HIPVs [sabinene, *n*-heptanal, α-pinene and (Z)-3-hexenyl acetate] (RC Treatt & Co. Ltd, Suffolk, UK; Wako Pure Chemical Industries, Ltd, Osaka, Japan; Tokyo Kasei Kogyo Co. Ltd, Tokyo, Japan) were mixed in a mass ratio of $1.8:1.3:2.0:3.0$ based on gas chromatographic analysis of emissions from infested plants [14]. The mixture, hereafter referred to as the 'attractant', was dissolved in triethyl citrate (TEC) to facilitate a low volatilization rate.

Dispensers were constructed by saturating each of five cellulose blocks ($22 \times 35 \times 2.8$ mm) with 1.7 g of TEC solution. These blocks were aligned in a row and wrapped in micropore polyethylene film ($30 \text{ mm} \times 40 \text{ mm} \times 100 \text{ μm}$) using a thermal sealing device to further ensure a low volatilization rate (for details of the dispenser, see [19]).

For carnivore-attracting HIPVs to be effective in the field, the concentrations used are of considerable importance. We previously reported that a bottle-type dispenser with 2 mg of a mixture of the four synthetic compounds in 20 ml of TEC attracted *C. vestalis* under field conditions [15]. During preliminary greenhouse experiments with the dispensers and feeders (see below) in the Miyama area, we used 4.25 mg attractant per dispenser (electronic supplementary material). The results indicated that this amount may not produce the concentrations required to be effective in an agricultural field in the Miyama area (electronic supplementary material, figure S1). Because the detection of HIPV concentrations is technically difficult in field experiments, we simulated dispersion with two different

amounts of attractant (4.25 mg and 425 mg) using a computational fluid dynamics approach [20]. Based on the results of this simulation and the preliminary field study, we decided to use a dispenser with 425 mg attractant for the greenhouse experiments (5% attractant in the TEC solution).

## 2.2. Greenhouse conditions

We conducted the field experiments in the Miyama rural area in Kyoto Prefecture, Japan (35.3° N, 135.5° E), where mizuna is one of the main greenhouse crops. Approximately 9000 mizuna plants were grown in each greenhouse (approximately 550 m$^3$). DBM larvae are among the most important pests of mizuna plants. DBM and their native larval parasitoid wasps *C. vestalis* occur on wild cruciferous weeds in the surrounding area, particularly on yellowcress plants (*Rorippa indica*, Brassicales: Brassicaceae) [18]. In the Miyama area, *C. vestalis* is the main parasitoid wasp of DBM larvae. We also observed *Oomyzus sokolowskii* (Hymenoptera: Eulophidae), a larval parasitoid wasp of DBMs, on rare occasions (J Abe 2003, unpublished data).

The farmers used prophylactic pesticides at the time of sowing to control the striped flea beetle, *Phyllotreta striolata* (Coleoptera: Chrysomelidae). Subsequently, the conditions inside the commercial greenhouses were controlled by each farmer, based on advice from the local agricultural experiment station. Some control measures were implemented upon detection of at least one DBM (either an adult or a larva) in a greenhouse. When farmers found DBM adults or larvae in their greenhouses, they removed them physically (manually or with a trapping device). Only when the densities of DBMs or other pest insects were so high that physical control was impossible did they use pesticides or solarization (i.e. abandoning the crop and using solar radiation to increase the internal greenhouse temperature to exterminate pest insects). Unfortunately, interviews with farmers did not provide a detailed history of pest control for each greenhouse except in the case of solarization. We excluded data from solarized greenhouses from the analyses.

The greenhouses used in this study were all covered with 1 mm nylon mesh to prevent the invasion of pest insects. However, slits around the entrance door and seams of the mesh allowed some adult DBMs and other pest insects to invade the greenhouses. Further, the procedure of covering each greenhouse was undertaken by the farmer, and consequently, the tightness of the mesh was not constant. Thus, we set treated and untreated greenhouses at each farmer's property to exclude any possible effects of differences in the tightness of greenhouse mesh; there was one exception to this pairing in each experimental year, owing to farmers using an allocated untreated greenhouse for a different purpose. It is unlikely that DBM larvae invaded the greenhouses from the surrounding area by crawling. *Cotesia vestalis* adults can pass through the 1 mm nylon mesh [15].

## 2.3. Experimental conditions

Dispensers of synthetic HIPVs that attract *C. vestalis* and honey feeders for *C. vestalis* were set inside greenhouses (treated greenhouses). The dispensers were placed every 5 m along two inside walls of each treated greenhouse. The dispensers were renewed every week in 2006 and every two weeks in 2008. In the greenhouses, no nectar or other sugary food was available for *C. vestalis*. Therefore, we placed feeders (one feeder per 100 m$^2$; 5 cm diameter × 7 cm tall glass bottles filled with honey; see [21] for details) along the inner wall of the treated greenhouses. The feeders were replaced with new ones every month. Neither dispensers nor feeders were placed in any of the untreated greenhouses. As the number of greenhouses belonging to each farmer varied from two to five, it was not possible to include additional treatment settings (e.g. only dispensers or only feeders) for each farmer.

To detect the occurrence of DBM adults and *C. vestalis*, we used a trapping method consisting of one yellow sticky trap-sheet (10 cm × 26 cm × 1 mm; Horiver, Arysta LifeScience Corporation, Tokyo, Japan) placed at the centre of each greenhouse. Because these were commercial farms, the number of trap-sheets was restricted to one per greenhouse at the request of the farmers who did not wish their activities to be impeded. The trap-sheets were replaced every week. From April to October in both years, we used greenhouses belonging to seven farmers in the Miyama area, 19 greenhouses in 2006 and 18 greenhouses in 2008. The total number of trap-sheets collected was 389 (178 in the untreated and 211 in the treated greenhouses) in 2006 and 415 (173 in the untreated and 242 in the treated greenhouses) in 2008.

The number of DBM adults and *C. vestalis* trapped on the sticky trap-sheets were counted using a stereoscopic microscope in the laboratory. DBM adults were identified by their characteristic morphology (the small, greyish-brown moth has a cream-coloured band that forms diamonds along

R. Soc. Open Sci. **7**: 201592

its back). For the identification of *C. vestalis*, the following two steps were used. First, we identified the genus of the trapped wasps by direct observation; when a wasp was identified as *Cotesia*, we carefully removed it from the sticky trap-sheet with fine forceps after applying a droplet of xylene. Next, we identified the species under a microscope based on Nixon [22] and Papp [23].

In the Miyama area, we detected not only DBM but also other pest insects of mizuna crops including several aphid species, striped flea beetles (*P. striolata*), vegetable weevils (*Listroderes costirostris*, Coleoptera: Curculionidae) and cabbage armyworms (*Mamestra brassicae*, Lepidoptera: Noctuidae). Some of these insects were occasionally trapped on the sticky trap-sheets; however, as our target insect pest was DBMs, any other insect species on the sheets were not counted. Hymenopterans other than *C. vestalis* were also not counted.

## 2.4. Data analyses

### 2.4.1. Comparisons between the treated and untreated greenhouses

As mentioned in §2.2, upon detection of at least one DBM (either adult or larva) in a greenhouse, some control measures (removing them manually or with a trapping device) were likely to be implemented by farmers, although exact details of control measures for each greenhouse were unavailable. Therefore, to compare the treated and untreated greenhouses, we did not compare the numbers of DBM and *C. vestalis* in the treated and untreated greenhouses at particular time points during the observation period. Instead, to minimize the effect of pest control by farmers, we compared the relative numbers of greenhouses per month in which more than one DBM adult (monthly incidence of DBM) or more than one *C. vestalis* (monthly incidence of *C. vestalis*) were found.

We analysed the effects of the treatment, month and their interaction on the monthly incidence of DBM in the treated and untreated greenhouses. We also analysed the effects of species (DBM or *C. vestalis*), month and their interaction on the monthly incidences of DBM and *C. vestalis* in the treated and untreated greenhouses. We used generalized linear mixed models (GLMMs) with a binomial distribution and logit-link, using the 'glmer' function in the 'lme4' package, v. 1.1–21 [24], in R v. 3.3.5 [25]. The greenhouse was a random effect in all models. Significant values from the GLMMs were calculated from type II Wald chi-square tests using the 'Anova' function in the 'car' package v. 3.0.2 [26]. In the GLMMs, the odds ratio (effect size) for each treatment factor, which estimated the effect of each treatment on the incidence of DBM while including greenhouse as a random effect, was calculated using the 'odds_to_rr' function in the 'sjstats' package, v. 0.18.0 [27], in R. In the event of convergence errors, the models were fitted using the 'bobyqa' optimizer in R.

### 2.4.2. The relationship between diamondback moth adults and *Cotesia vestalis* in the treated and untreated greenhouses

To analyse the relationship between DBM adults and *C. vestalis* in the treated and untreated greenhouses, we focused on the number of *C. vestalis* in relation to the increased number of DBM throughout the observation period. To do this, we used Poisson regression analysis in JMP v. 14.2.0 (SAS Institute, 2018). The numbers of DBMs and *C. vestalis* in each greenhouse at particular time points during the observation period were plotted.

# 3. Results

## 3.1. Monthly incidence of adult diamondback moths in the treated and untreated greenhouses

First, we compared the monthly incidence of DBMs between the treated and untreated greenhouses (figure 1: white bars versus grey bars in each year). In both 2006 (table 1*a*) and 2008 (table 1*b*), DBM incidence was significantly affected by treatment ($p = 0.0149$ and $0.0388$, respectively) and month ($p = 0.0243$ and $0.0024$, respectively), but not by their interaction (treatment × month) ($p = 0.1357$ and $0.6970$, respectively). As the interaction was not significant in either year (table 1), we compared the pooled data of the monthly incidence of adult DBMs between the treated and untreated greenhouses. In both years, the incidence of DBM in the treated greenhouses was significantly lower than that in the untreated greenhouses (figure 2). The odds ratios in 2006 and 2008 were 0.3339 and 0.2375, respectively.

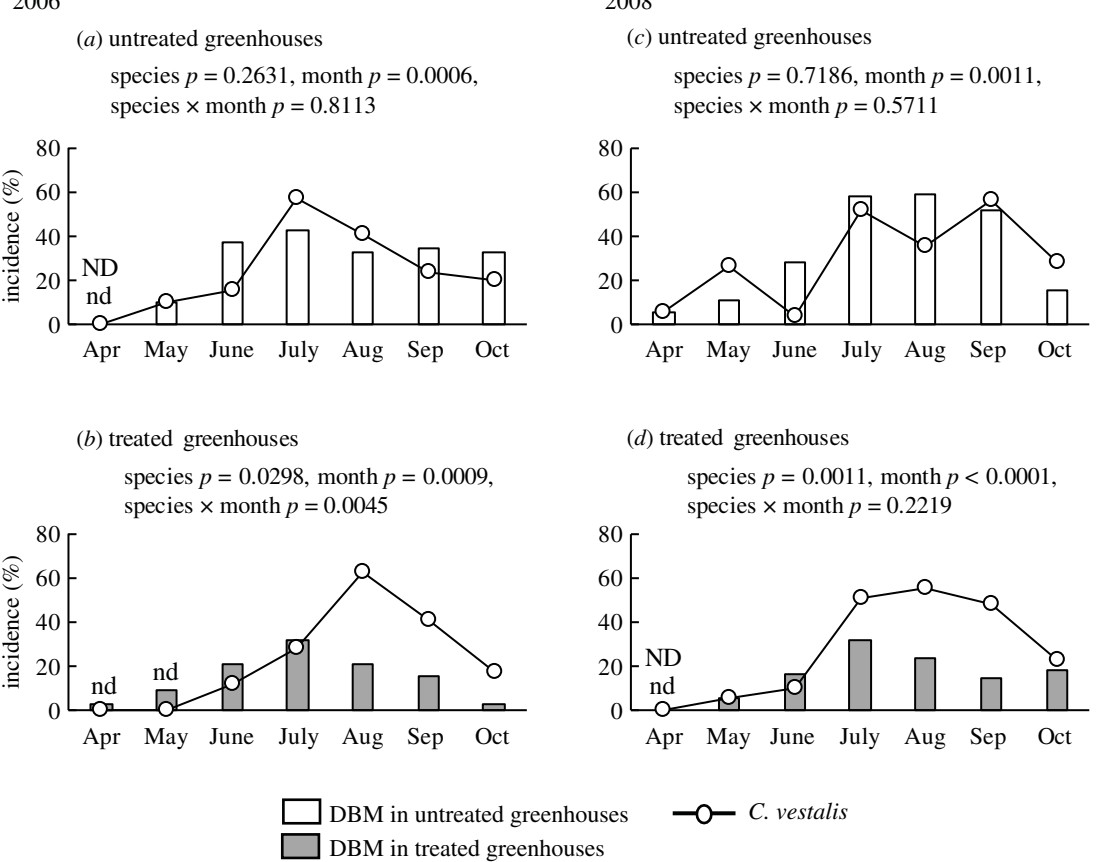

**Figure 1.** Comparison between the relative numbers of greenhouses in which more than one diamondback moth or more than one *C. vestalis* was detected per month. ND: Diamondback moths were not detected; nd: *C. vestalis* were not detected.

**Table 1.** Results from generalized linear mixed models (GLMMs) analysing the effects of the treatment, month and their interaction on the monthly incidence of diamondback moths in 2006 and 2008.

| effects | $\chi^2$ | d.f. | $p$ |
|---|---|---|---|
| (a) 2006 | | | |
| treatment | 5.9323 | 1 | 0.0149 |
| month | 5.0719 | 1 | 0.0243 |
| treatment × month | 2.2265 | 1 | 0.1357 |
| (b) 2008 | | | |
| treatment | 4.2718 | 1 | 0.0388 |
| month | 9.1974 | 1 | 0.0024 |
| treatment × month | 0.1516 | 1 | 0.6970 |

## 3.2. Monthly incidences of adult diamondback moths and *Cotesia vestalis*

Next, we compared the monthly incidences of DBM and *C. vestalis* (lines versus bars in the same subfigures). In 2006, in the untreated greenhouses, the incidences of DBM and *C. vestalis* were not significantly affected by species ($p = 0.2631$) or the species–month interaction ($p = 0.8113$) (white bars versus solid line in figure 1$a$, table 2$a$); however, they were significantly affected by month ($p = 0.0006$) (figure 1$a$, table 2$a$). By contrast, in the treated greenhouses, the incidences were significantly affected by species ($p = 0.0298$), month ($p = 0.0009$), and the species–month interaction ($p = 0.0045$) (grey bars versus solid line in figure 1$b$, table 2$b$).

Similar trends were detected in 2008. In the untreated greenhouses, the incidences were not significantly affected by species ($p = 0.7186$) or the species–month interaction ($p = 0.5711$) (white bars versus solid line in

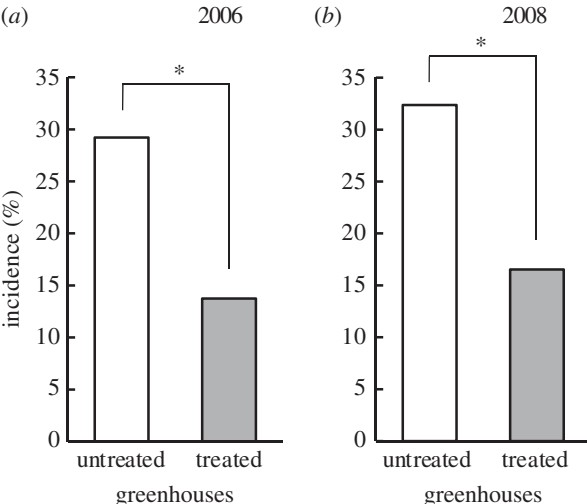

**Figure 2.** Pooled data of the incidence of adult diamondback moths in the treated and untreated greenhouses in 2006 and 2008. Asterisks indicate a significant difference (*: $0.05 > p > 0.01$). For detailed *p*-values, table 1, treatment factors).

**Table 2.** Results from generalized linear mixed models (GLMMs) analysing the effects of the species (DBM and *C. vestalis*), month and their interaction on the monthly incidence of diamondback moths and *C. vestalis* in 2006 and 2008.

| effects | $\chi^2$ | d.f. | $p$ |
|---|---|---|---|
| (a) 2006 | | | |
| untreated greenhouses | | | |
| species | 1.2526 | 1 | 0.2631 |
| month | 11.7256 | 1 | 0.0006 |
| species × month | 0.0570 | 1 | 0.8113 |
| treated greenhouses | | | |
| species | 4.7222 | 1 | 0.0298 |
| month | 11.0812 | 1 | 0.0009 |
| species × month | 8.0798 | 1 | 0.0045 |
| (b) 2008 | | | |
| control greenhouses | | | |
| species | 0.1299 | 1 | 0.7186 |
| month | 10.7370 | 1 | 0.0011 |
| species × month | 0.3209 | 1 | 0.5711 |
| treated greenhouses | | | |
| species | 10.6925 | 1 | 0.0011 |
| month | 20.8151 | 1 | <0.0001 |
| species × month | 1.4918 | 1 | 0.2219 |

figure 1*c*) (table 2*b*), but were significantly affected by month ($p = 0.0011$) (figure 1*c*, table 2*b*). By contrast, in the treated greenhouses, the incidences were significantly affected by species ($p = 0.0011$) and month ($p < 0.0001$) (grey bars versus solid line in figure 1*d*, table 2*b*). However, unlike 2006, they were not significantly affected by the species–month interaction ($p = 0.2219$) (figure 1*d*, table 2*b*).

## 3.3. The relationship between diamondback moth adults and *Cotesia vestalis* in the treated and untreated greenhouses

In both 2006 and 2008, the number of DBM adults had a significant effect on the number of *C. vestalis* adults, while treatment alone had no significant effect (table 3*a,b*, figure 3*a,b*). The interaction

**Table 3.** Results from Poisson regression analyses assessing the effects of the treatment, the diamondback moth number (DBM) and their interaction on the *C. vestalis* number in 2006 and 2008.

| effects | $\chi^2$ | d.f. | $p$ |
|---|---|---|---|
| (*a*) 2006 | | | |
| treatment | 0.9269 | 1 | 0.3557 |
| DBM | 22.4950 | 1 | <0.0001 |
| treatment × DBM | 10.8468 | 1 | 0.0010 |
| (*b*) 2008 | | | |
| treatment | 0.1655 | 1 | 0.6842 |
| DBM | 359.5024 | 1 | <0.0001 |
| treatment × DBM | 45.6222 | 1 | <0.0001 |

(treatment × DBM number) significantly affected the number of *C. vestalis* adults in both years (table 3*a*,*b*, figure 3).

## 4. Discussion

We confirmed that the attractant did not affect the oviposition behaviour of DBM adults on mizuna plants (the number of eggs), the pupation rate and the pupal weight of DBM larvae on mizuna plants (electronic supplementary materials and figure S2). Thus, in contrast with other studies [28–34], the attractant did not affect the performance of either DBM adults or larvae. We could therefore focus solely on the *C. vestalis*-attracting function of the HIPV dispensers.

The monthly incidence of DBMs in the greenhouses was influenced by two factors: (i) DBM adults that invade from the surroundings by chance (i.e. the invaders = the first generation), and (ii) DBM adults that emerged in the greenhouses (second or later generations). Although the greenhouses used in this study were covered with 1 mm nylon mesh, there were slits that facilitated the entry of DBM adults into the greenhouses. The probability of such chance invasions was expected to be small and similar between the treated and untreated greenhouses, both of which were set in a farm field. Thus, because the incidences of DBM in the treated greenhouses were significantly lower than those in the untreated greenhouses across the 2-year period (figure 1: white bars versus grey bars in each year, and figure 2) and the odds ratios, which indicates the relative likelihood of the incidence of DBM, in the treated greenhouses were lower than those in the untreated greenhouses in both years (2006: 0.3339; 2008: 0.2375), we concluded that the occurrence of second or later generations of DBM larvae was suppressed more effectively in the treated greenhouses than in the untreated greenhouses.

In the treated greenhouses, the monthly incidences of *C. vestalis* were significantly higher than those of DBM adults in both years (figure 1*b*,*d*). However, such significant differences were not observed in the untreated greenhouses in either year (figure 1*a*,*c*). Together with our previous greenhouse/field studies showing that *C. vestalis* females are attracted to the attractant under both laboratory and field conditions [15–20], we inferred that the significant difference in the treated greenhouses could be attributed to the artificial attraction of *C. vestalis* from the surrounding area by the dispensers, irrespective of the presence or absence of DBM adults in the treated greenhouses. Another explanation for the results of figure 1 (lines versus bars in the same subfigures) is that the difference was caused not by the dispensers but by the honey feeders: both the treated and untreated greenhouses were visited equally by *C. vestalis* irrespective of the presence or absence of dispensers in the greenhouses, but the wasps tended to stay longer and parasitized more DBM larvae in the treated greenhouses where there were honey feeders. However, we believe this explanation is less plausible because, in our preliminary experiments using feeders and dispensers containing lower amounts of HIPVs (1/100) installed according to the same criteria used in 2006 and 2008 (electronic supplementary materials), the monthly incidences of DBM in the treated and untreated greenhouses were not significantly different (electronic supplementary material, figure S1).

In the Poisson regression analyses (figure 3), it is important to note that the interaction (treatment × DBM number) was significant in both years, with a higher number of *C. vestalis* recorded in the treated

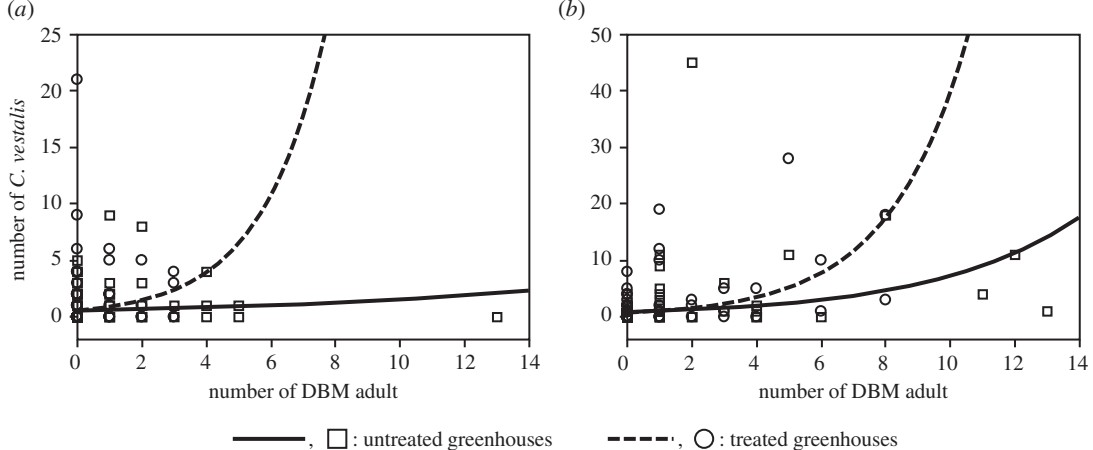

Figure 3. The relationship between the numbers of diamondback moth and *C. vestalis* using Poisson regression. (*a*) 2006; (*b*) 2008.

greenhouses than in the untreated ones, when there was an increase in the number of DBM adults. Thus, the significant interaction observed in both years further supports the effectiveness of the tested treatment in greenhouses.

# 5. Conclusion

These experiments demonstrate the potential for successful conservation biological control of DBMs using HIPVs that attract specified native natural enemies (*C. vestalis*) in conjunction with feeders. We anticipate that, by adopting this method, farmers may be able to reduce their use of pesticides for DBMs in greenhouses.

In this study, we used 18–19 greenhouses belonging to seven farmers in two separate years (2006 and 2008). To test the effectiveness of this method for conservation biological control, it is important to extend the research to include more greenhouses across consecutive years. Such experiments would ascertain the resilience of the DBM and *C. vestalis* populations in the surrounding areas to this method.

To ensure a viable population of *C. vestalis*, it is important to investigate an alternative local host species. The larvae of *Leuroperna sera* (Lepidoptella: Putellidae) are alternative hosts of *C. vestalis* which live in Miyama and have the added benefit of not being pests in greenhouses (Abe 2004, personal observation); the presence of this species would affect the populations of DBM and *C. vestalis* in the surrounding areas and, therefore, clearly warrants further study.

Furthermore, we plan to study whether the strategy of using synthetic HIPVs dispensers and feeders for native *C. vestalis* may be used as an effective means of controlling DBMs in open agricultural fields.

Data accessibility. Electronic supplementary data are available online at the Dryad Digital Repository: https://doi.org/10.5061/dryad.fqz612jpq [35].

Authors' contributions. M.U. and J.A. contributed equally to this work. M.U., J.A. and K.S. designed and conducted the experiments, and analysed the data. J.T. designed the experiments, analysed the data and wrote the manuscript. S.U. and K.N. participated in the design of the study. All authors gave final approval of the manuscript for publication.

Competing interests. The authors declare that they have no known competing financial interests or personal relationships that could have appeared to influence the work reported in this paper.

Funding. This study was supported by the Bio-oriented Technology Research Advancement Institution from the Ministry of Agriculture, Forestry and Fisheries, by Science and Engineering Entrepreneurship Development Program for Vigorous Researchers (SEED-V) from Japan Science and Technology Agency and by grants for the scientific research of a priority area (S) [grant no. 19101009] and scientific research (B) [grant no. 26292030] from the Ministry of Education, Culture, Sports, Science and Technology.

Acknowledgements. We thank the members of the project funded by Bio-oriented Technology Research Advancement Institution for their helpful discussions; S. Kugimiya, T. Shimoda and Y. Nakashima for their comments on the early version of the manuscript, and K. Sano, H. Oyagi and N. Nose for their support with field experiments. We thank Editage (www.editage.com) for English language editing.

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
