## [Reviewer comments · Royal Society Open Science]

Review History

RSOS-200184.R0 (Original submission)

Review form: Reviewer 1

Is the manuscript scientifically sound in its present form?

Yes

Are the interpretations and conclusions justified by the results?

Yes

Is the language acceptable?

Yes

Do you have any ethical concerns with this paper?

No

Have you any concerns about statistical analyses in this paper?

Yes

Recommendation?

Accept with minor revision (please list in comments)

Comments to the Author(s)

Comments for Uefune et al. (Royal Society Open Science)

This study documents improved control of diamondback moths in greenhouses that have been treated with synthetic VOCs similar to those emitted by infected crucifers. In the past, growers have had more successes with biocontrol efforts using natural enemies released in greenhouses than in field situations and it is exciting to include VOC signaling in these efforts. The experimental design was simple and the results were convincing and novel.

1. I recommend that you add analyses comparing the relationship between DBM and *C. v.* in treated and control greenhouses. In control greenhouses, numbers of *C. v.* seem to follow those of DBM while in the treated greenhouses, numbers of parasitoids remained high later in the season even when DBM became less common. This is mentioned in the summary and alluded to in section 4.3. A statistical analysis would be helpful (regression or correlation) with treatment included. A figure showing the fit between these two would also be helpful.
2. Is it possible that some of the difference that you observed was caused by the honey and not by the attractant? Perhaps both treatments attracted parasitoids but they only stayed where there was honey and left where there was none. This may be unlikely but it may be worth mentioning. If it seems unlikely, explain why. I had this question particularly when I read L 231-235.
3. You measured % of greenhouses with more than 1 DBM or *C. v.* This seems like an odd threshold. Are results qualitatively similar if other measures of DBM or *C. v.* are used as the threshold?
4. Please give some estimate of the effect size, in addition to the P values. For instance, DBM were found in approximately half the treated greenhouses compared to controls.
5. These experiments were conducted over a relatively small temporal and spatial scale. I wonder if this technique will continue to be effective over longer time frames and if more greenhouses were involved. What will happen if the VOCs continue to attract parasitoids but there are not enough DBM hosts? Will there always be enough hosts in the surrounding fields to augment populations of *C. v.* adults?

This paper was very well written. I have several suggestions for improving the presentation:

L 48-49 I agree that emission of HIPVs provides defense but they may have other functions as well. This wording implies that they have evolved to provide defense. I would tone this down a bit – add a few words like may provide defense.

L 68-69 a blend has been shown

L 84 Add more - gas chromatographic analysis of emissions from infected plants?

L 87-88 cardboard means a paper product – corrugated cellulose?

L 111 In the Miyama area

L 112 delete and

L 238 Delete this first sentence and start with: These experiments ...

L 239-243 This is hard to follow – make this two sentences.

Review form: Reviewer 2

Is the manuscript scientifically sound in its present form?

No

Are the interpretations and conclusions justified by the results?

No

Is the language acceptable?

No

Do you have any ethical concerns with this paper?

No

Have you any concerns about statistical analyses in this paper?

Yes

Recommendation?

Reject

Comments to the Author(s)

This paper examines the possibility of using synthetic H1VP lures to increase the impact of parasitoids on populations of the diamondback moth (BBM) in greenhouses. This is an interesting idea and the results suggest that pest populations are lower in the experimental compared with control greenhouses.

However, there are a number of aspects that need to be addressed before the paper should be considered for publication.

Looking at Figure 2 it would appear that the proportion of greenhouses with parasitoids did not differ a lot between treatment and controls. Thus, one has to ask if the difference was actually due to the presence of a food source in the experimental greenhouses; with food the parasitoids could be living longer and thus attacking a higher proportion of larvae? This seems rather likely as DBM eggs etc (see supplements) were not different.

While there appear to be 9 treatment and 9 control greenhouses each year (although in 2006 there appears to be an extra control) it is unclear how many farms were used. As there were up to five greenhouses on one farm, there could be two replicates at the same site. If that was the case should site not also be included in the analysis, as the levels of infestation could vary between farms.

The actual sampling, using one sticky trap per greenhouse, especially as it was placed in the middle when insects would most likely enter at the edges, seems rather minimal. Furthermore, just the presence or absence of one individual per month does not really reflect population dynamics.

From a presentation perspective the manuscript needs to be edited both for content and for English. With respect to content, the information on other species observed (start of results) really adds nothing. Also, Figure 1 seems somewhat unnecessary as the data are also in Figure 2. Throughout rate is used when in fact it is incidence, as there is no time function associated with the data.

Decision letter (RSOS-200184.R0)

27-Feb-2020

Dear Dr Takabayashi:

Manuscript ID RSOS-200184 entitled "Targeting diamondback moth in greenhouses by attracting specific native parasitoids with herbivory-induced plant volatiles" which you submitted to Royal Society Open Science, has been reviewed. The comments from reviewers are included at the bottom of this letter.

In view of the criticisms of the reviewers, the manuscript has been rejected in its current form. However, a new manuscript may be submitted which takes into consideration these comments.

Please note that resubmitting your manuscript does not guarantee eventual acceptance, and that your resubmission will be subject to peer review before a decision is made.

Your resubmitted manuscript should be submitted by 26-Aug-2020. If you are unable to submit by this date please contact the Editorial Office.

on behalf of Dr Richard Benton (Associate Editor) and Pete Smith (Subject Editor)
openscience@royalsociety.org

Reviewers' Comments to Author:
Reviewer: 1

Comments to the Author(s)
Comments for Uefune et al. (Royal Society Open Science)

This study documents improved control of diamondback moths in greenhouses that have been treated with synthetic VOCs similar to those emitted by infected crucifers. In the past, growers have had more successes with biocontrol efforts using natural enemies released in greenhouses than in field situations and it is exciting to include VOC signaling in these efforts. The experimental design was simple and the results were convincing and novel.

1. I recommend that you add analyses comparing the relationship between DBM and *C. v.*

in treated and control greenhouses. In control greenhouses, numbers of *C. v.* seem to follow those of DBM while in the treated greenhouses, numbers of parasitoids remained high later in the season even when DBM became less common. This is mentioned in the summary and alluded to in section 4.3. A statistical analysis would be helpful (regression or correlation) with treatment included. A figure showing the fit between these two would also be helpful.

2. Is it possible that some of the difference that you observed was caused by the honey and not by the attractant? Perhaps both treatments attracted parasitoids but they only stayed where there was honey and left where there was none. This may be unlikely but it may be worth mentioning. If it seems unlikely, explain why. I had this question particularly when I read L 231-235.
3. You measured % of greenhouses with more than 1 DBM or *C. v.* This seems like an odd threshold. Are results qualitatively similar if other measures of DBM or *C. v.* are used as the threshold?
4. Please give some estimate of the effect size, in addition to the P values. For instance, DBM were found in approximately half the treated greenhouses compared to controls.
5. These experiments were conducted over a relatively small temporal and spatial scale. I wonder if this technique will continue to be effective over longer time frames and if more greenhouses were involved. What will happen if the VOCs continue to attract parasitoids but there are not enough DBM hosts? Will there always be enough hosts in the surrounding fields to augment populations of *C. v.* adults?

This paper was very well written. I have several suggestions for improving the presentation:

L 48-49 I agree that emission of HIPVs provides defense but they may have other functions as well. This wording implies that they have evolved to provide defense. I would tone this down a bit – add a few words like may provide defense.

L 68-69 a blend has been shown

L 84 Add more - gas chromatographic analysis of emissions from infected plants?

L 87-88 cardboard means a paper product – corrugated cellulose?

L 111 In the Miyama area

L 112 delete and

L 238 Delete this first sentence and start with: These experiments ...

L 239-243 This is hard to follow – make this two sentences.

Reviewer: 2

Comments to the Author(s)

This paper examines the possibility of using synthetic HVP lures to increase the impact of parasitoids on populations of the diamondback moth (BBM) in greenhouses. This is an interesting idea and the results suggest that pest populations are lower in the experimental compared with control greenhouses.

However, there are a number of aspects that need to be addressed before the paper should be considered for publication.

Looking at Figure 2 it would appear that the proportion of greenhouses with parasitoids did not differ a lot between treatment and controls. Thus, one has to ask if the difference was actually due to the presence of a food source in the experimental greenhouses; with food the parasitoids could be living longer and thus attacking a higher proportion of larvae? This seems rather likely as DBM eggs etc (see supplements) were not different.

While there appear to be 9 treatment and 9 control greenhouses each year (although in 2006 there appears to be an extra control) It is unclear how many farms were used. As there were up to five greenhouses on one farm, there could be two replicates at the same site. If that was the case should site not also be included in the analysis, as the levels of infestation could vary between farms.

The actual sampling, using one sticky trap per greenhouse, especially as it was placed in the middle when insects would most likely enter at the edges, seems rather minimal. Furthermore, just the presence or absence of one individual per month does not really reflect population dynamics.

From a presentation perspective the manuscript needs to be edited both for content and for English. With respect to content, the information on other species observed (start of results) really adds nothing. Also, Figure 1 seems somewhat unnecessary as the data are also in Figure 2. Throughout rate is used when in fact it is incidence, as there is no time function associated with the data.

Author's Response to Decision Letter for (RSOS-200184.R0)

See Appendix A.

RSOS-201592.R0

Review form: Reviewer 1

Is the manuscript scientifically sound in its present form?

Yes

Are the interpretations and conclusions justified by the results?

Yes

Is the language acceptable?

Yes

Do you have any ethical concerns with this paper?

No

Have you any concerns about statistical analyses in this paper?

No

Recommendation?

Accept as is

Comments to the Author(s)

The authors have done a nice job of addressing my concerns. I recommend that the manuscript be accepted for publication.

Review form: Reviewer 2**Is the manuscript scientifically sound in its present form?**

Yes

Are the interpretations and conclusions justified by the results?

Yes

Is the language acceptable?

Yes

Do you have any ethical concerns with this paper?

No

Have you any concerns about statistical analyses in this paper?

No

Recommendation?

Major revision is needed (please make suggestions in comments)

Comments to the Author(s)

From a technical perspective I believe they authors have addressed the reviewers' comments in a satisfactory manner, and feel the information certainly merits publication.

However, the text would benefit from editing for presentation. For example, in the first paragraph of the results lines 255-261 could be " In both 2006 (Table 1A) and 2008 (Table 1B) the incidence of DBM was significantly affected by treatment and month, but there were no significant treatment x month interactions"

Decision letter (RSOS-201592.R0)

Dear Dr Takabayashi

On behalf of the Editors, we are pleased to inform you that your Manuscript RSOS-201592 "Targeting diamondback moth in greenhouses by attracting specific native parasitoids with herbivory-induced plant volatiles" has been accepted for publication in Royal Society Open Science subject to minor revision in accordance with the referees' reports. Please find the referees' comments along with any feedback from the Editors below my signature.

We invite you to respond to the comments and revise your manuscript. Below the referees' and Editors' comments (where applicable) we provide additional requirements. Final acceptance of

your manuscript is dependent on these requirements being met. We provide guidance below to help you prepare your revision.

Please submit your revised manuscript and required files (see below) no later than 7 days from today's (ie 14-Oct-2020) date. Note: the ScholarOne system will 'lock' if submission of the revision is attempted 7 or more days after the deadline. If you do not think you will be able to meet this deadline please contact the editorial office immediately.

on behalf of Dr Richard Benton (Associate Editor) and Pete Smith (Subject Editor)
openscience@royalsociety.org

Reviewer comments to Author:
Reviewer: 1

Comments to the Author(s)
The authors have done a nice job of addressing my concerns. I recommend that the manuscript be accepted for publication.

Reviewer: 2

Comments to the Author(s)
From a technical perspective I believe they authors have addressed the reviewers' comments in a satisfactory manner, and feel the information certainly merits publication.

However, the text would benefit from editing for presentation. For example, in the first paragraph of the results lines 255-261 could be " In both 2006 (Table 1A) and 2008 (Table 1B) the incidence of DBM was significantly affected by treatment and month, but there were no significant treatment x month interactions"

===PREPARING YOUR MANUSCRIPT===

- one version identifying all the changes that have been made (for instance, in coloured highlight, in bold text, or tracked changes);
- a 'clean' version of the new manuscript that incorporates the changes made, but does not highlight them. This version will be used for typesetting.

===PREPARING YOUR REVISION IN SCHOLARONE===

-- Ensure that your data access statement meets the requirements at <https://royalsociety.org/journals/authors/author-guidelines/#data>. You should ensure that you cite the dataset in your reference list. If you have deposited data etc in the Dryad repository, please only include the 'For publication' link at this stage. You should remove the 'For review' link.

Author's Response to Decision Letter for (RSOS-201592.R0)

See Appendix B.

Decision letter (RSOS-201592.R1)

Dear Dr Takabayashi,

It is a pleasure to accept your manuscript entitled "Targeting diamondback moths in greenhouses by attracting specific native parasitoids with herbivory-induced plant volatiles" in its current form for publication in Royal Society Open Science.

on behalf of the Associate Editor and Professor Pete Smith (Subject Editor)
openscience@royalsociety.org

Appendix A

Dear the editor and reviewers

The followings are the point-by-point responses to the comments by the reviewers. The line numbers are based on the file “RSOS (accept all changed and stop tracking)”, because the line numbers of the file RSOS resubmit (track change) were quite confusing.

Reviewer: 1

Comment 1.

I recommend that you add analyses comparing the relationship between DBM and *C. v.* in treated and control greenhouses. In control greenhouses, numbers of *C. v.* seem to follow those of DBM while in the treated greenhouses, numbers of parasitoids remained high later in the season even when DBM became less common. This is mentioned in the summary and alluded to in section 4.3. A statistical analysis would be helpful (regression or correlation) with treatment included. A figure showing the fit between these two would also be helpful.

Response 1.

Thank you for your comment. We have added a new figure showing the results of a Poisson regression analysis on the relationship between DBM and *C. vestalis* in treated and control greenhouses throughout the observation period (this is labelled as Figure 2; please note that Figure 1 of the previous version has been deleted based on a comment from reviewer 2), and we discuss the effectiveness of the treatment.

The following text has been added:

Lines 208-215 (Materials and Methods)

“3.4.2. The relationship between DBM adults and *C. vestalis* within either treated or untreated greenhouses.

To analyse the relationship between DBM adults and *C. vestalis* in treated and untreated greenhouses, we focused on the number of *C. vestalis* in relation to the increased number of DBM throughout the observation period. To do this, we used a Poisson regression analysis in JMP version 14.2.0 (SAS Institute, 2018). The numbers of DBM and *C. vestalis* in a greenhouse at particular times of the observation period were plotted.”

Lines 250-255 (Results)

“4.3. The relationship between DBM adults and *C. vestalis* in the treated and untreated greenhouses In both 2006 and 2008, only the number of DBM adults had a significant effect on the number of *C. vestalis* adults, while treatment alone had no significant effect (Table 3A, B) (Fig. 3A, B). The interaction (treatment \times DBM number) significantly affected the number of *C. vestalis* adults in both years (Table 3A, B) (Fig. 3).”

Lines 298-303 (Discussion)

“In the Poisson regression analyses (Fig. 3), it is important to note that the interaction (treatment × DBM number) was significant in both years, with a higher number of *C. vestalis* recorded in the treated greenhouses than in the untreated ones, where there was an increase in the number of DBM adults. Thus, the significant interaction observed in both years further supports the effectiveness of the treatment in greenhouses.”

Comment 2.

Is it possible that some of the difference that you observed was caused by the honey and not by the attractant? Perhaps both treatments attracted parasitoids but they only stayed where there was honey and left where there was none. This may be unlikely but it may be worth mentioning. If it seems unlikely, explain why. I had this question particularly when I read L 231-235.

Response 2.

We mentioned this possibility in lines 261-265 of the original version. Here, we discussed that this explanation was unlikely by using the results of our preliminary experiments, as mentioned in lines 96-100. We have clarified this in the revised version, where the results of the preliminary experiments are shown as Supplemental Figure 1. We have also added the following text:

Lines 287-297

“Another explanation for the results of Fig. 1 (lines vs. bars in the same subfigures) is that the difference was caused not by the dispensers but by the honey feeders: both the treated and untreated greenhouses were visited equally by *C. vestalis* irrespective of the presence or absence of dispensers in the greenhouses, and the wasps tended to stay longer and parasitized more DBM larvae in the treated greenhouses where there were honey feeders. However, we believe this is less plausible, because in our preliminary experiments, we used dispensers with a lower amount of HIPVs (1/100) and the feeders, both set according to the same criteria as used in 2006 and 2008 (Supplementary materials), and the results showed that the monthly incidences of DBM in the treated and untreated greenhouses were not significantly different (Supplementary Fig. 1).”

Comment 3.

You measured % of greenhouses with more than 1 DBM or *C. v.* This seems like an odd threshold. Are results qualitatively similar if other measures of DBM or *C. v.* are used as the threshold?

Response 3.

The farmers' policy to control DBM is that control measures are implemented upon detection of at least one DBM (either an adult or a larva) in a greenhouse. Hence, we based our threshold upon this

farming practice, and measured percentage of greenhouses with more than 1 DBM or *C. vestalis*. We have now explained this in two parts of the methods section: lines 118-130 and lines 185-193.

Lines 118-130

“Farmers used prophylactic pesticides at the time of sowing to control the striped flea beetle, *Phyllotreta striolata*. Subsequently, the conditions inside the commercial greenhouses where this study was conducted were controlled by each farmer, based on advice from the local agricultural experiment station, with some control measure implemented upon detection of at least one DBM (either an adult or a larva) in a greenhouse. When farmers found DBM adults or larvae in their greenhouses, they removed them physically (manually or with a trapping device). Only when the densities of DBM or other pest insects were so high as to make physical control impossible did they use pesticides or solarization (i.e., abandoning the crop and using solar radiation to make the internal greenhouse temperature high enough to exterminate pest insects). Unfortunately, interviews with farmers did not provide a detailed history of pest control for each greenhouse except in the case of solarization. We excluded the data from the solarized greenhouses from the analyses.”

Lines 185-193

“As mentioned in 3.2, upon detection of at least one DBM (either adult or larva) in a greenhouse, some control measures (removing them manually or with a trapping device) were likely to be implemented by farmers, although exact details of control measures for each greenhouse were unavailable. Therefore, to compare the treated and untreated greenhouses, we did not use the numbers of DBM and *C. vestalis* in the treated and untreated greenhouses at a particular time in the observation period. Rather, to minimize the effect of pest control by farmers, we compared the relative numbers of greenhouses per month in which more than one DBM adult (monthly incidence of DBM) or more than one *C. vestalis* (monthly incidence of *C. vestalis*) were found.”

Comment 4.

Please give some estimate of the effect size, in addition to the P values. For instance, DBM were found in approximately half the treated greenhouses compared to controls.

Response 4.

We calculated the effect size (Odds ratio) for the treatment and added them in Figure 2.

Further, as the interactions (treatment X month) in both treated and control greenhouses in two years were not significant (Table 1), we added the pooled data of the monthly incidence of DBM and made a new figure (Figure 2 in the revised version) to clearly show the effects of the treatment on the incidence of DBM.

Comment 5.

These experiments were conducted over a relatively small temporal and spatial scale. I wonder if this technique will continue to be effective over longer time frames and if more greenhouses were involved. What will happen if the VOCs continue to attract parasitoids but there are not enough DBM hosts? Will there always be enough hosts in the surrounding fields to augment populations of *C. v.* adults?

Response 5.

We have added several sentences in response to these comments in the conclusion section.

Lines 310-320

“In this study, we used 18-19 greenhouses from seven farmers in two separate years (2006 and 2008). To test the effectiveness of this method for conservation biological control, it is important to extend the research to include more greenhouses across consecutive years. Such experiments would ascertain the resilience of DBM and *C. vestalis* populations in the surrounding areas to this method. To ensure a viable population of *C. vestalis*, it is important to investigate an alternative local host species. The larvae of *Leuoperna sera* (Lepidoptella, Putellidae) are hosts of *C. vestalis* which live in Miyama and have the added benefit of not being pests in greenhouses (Abe, personal observation); the presence of this species would affect the populations of DBM and *C. vestalis* in the surrounding areas, so it clearly warrants further study.”

Comment 6.

L 48-49 I agree that emission of HIPVs provides defense but they may have other functions as well. This wording implies that they have evolved to provide defense. I would tone this down a bit – add a few words like may provide defense.

Response 6.

We have changed the wording in accordance with this comment.

Comment 7.

L 68-69 a blend has been shown

Response 7.

We have revised this sentence.

Comment 8.

L 84 Add more - gas chromatographic analysis of emissions from infected plants?

Response 8.

We have revised this sentence.

Comment 9.

L 87-88 cardboard means a paper product – corrugated cellulose?

Response 9.

It was indeed cellulose, but not actually corrugated, so we have changed this wording to "cellulose block", i.e. "five cellulose blocks (22 × 35 × 2.8 mm) with..."

Comment 10.

L 111 In the Miyama area

Response 10.

We have revised this sentence.

Comment 11.

L 112 delete and

Response 11.

We have revised this sentence.

Comment 12.

L 238 Delete this first sentence and start with: These experiments ...

Response 12.

We have revised this sentence.

Comment 13.

L 239-243 This is hard to follow – make this two sentences.

Response 13.

We have split the sentence into two (lines 306-309):

“These experiments demonstrated the potential for successful conservation biological control of DBM by using HIPVs that attract specified native natural enemies (*C. vestalis*) in conjunction with feeders. We anticipate that, by adopting this method, farmers may be able to reduce their use of pesticides for DBM in greenhouses.”

Reviewer: 2

Comment 1.

Looking at Figure 2 it would appear that the proportion of greenhouses with parasitoids did not differ a lot between treatment and controls. Thus, one has to ask if the difference was actually due to the presence of a food source in the experimental greenhouses; with food the parasitoids could be

living longer and thus attacking a higher proportion of larvae? This seems rather likely as DBM eggs etc (see supplements) were not different.

Response 1.

This viewpoint, together with that expressed by Reviewer 1, are now included within the Discussion section (lines 287-297). We also appraise these viewpoints by referring to the results of the preliminary field experiment, as mentioned in the Materials and Methods section (lines 96-106). In the revised version, we have also included the results of the preliminary experiments as Supplemental Figure 1.

Lines 287-297 (Discussion)

“Another explanation for the results of Fig. 1 (lines vs. bars in the same subfigures) is that the difference was caused not by the dispensers but by the honey feeders: both the treated and untreated greenhouses were visited equally by *C. vestalis* irrespective of the presence or absence of dispensers in the greenhouses, and the wasps tended to stay longer and parasitized more DBM larvae in the treated greenhouses where there were honey feeders. However, we believe this is less plausible, because in our preliminary experiments, we used dispensers with a lower amount of HIPVs (1/100) and the feeders, both set according to the same criteria as used in 2006 and 2008 (Supplementary materials), and the results showed that the monthly incidences of DBM in the treated and untreated greenhouses were not significantly different (Supplementary Fig. 1).”

Comment 2.

While there appear to be 9 treatment and 9 control greenhouses each year (although in 2006 there appears to be an extra control) It is unclear how many farms were used.

As there were up to five greenhouses on one farm, there could two replicates at the same site. If that was the case should site not also be included in the analysis, as the levels of infestation could vary between farms.

Response 2.

Seven farmers joined this research (Line 161). In the case of using five greenhouses, the treated and control greenhouses were not paired, because individual greenhouses had different history of the pest-control measures done by farmers. Thus, in the model, the greenhouse was set as a random effect.

Comment 3.

The actual sampling, using one sticky trap per greenhouse, especially as it was placed in the middle when insects would most likely enter at the edges, seems rather minimal.

Response 3.

We were only allowed to put one trap at the centre of the greenhouses, because the farmers said that the presence of multiple traps would hamper their activities. Thus, we had to make a pragmatic choice in order for these experiments to proceed. But we aim to increase sampling levels in future projects, now that we can present the evidence from this study to farmers, to demonstrate how our approach has the potential to improve their pest control strategies.

Comment 4.

Furthermore, just the presence or absence of one individual per month does not really reflect population dynamics.

Response 4.

The farmers' policy to control DBM is that control measures are implemented upon detection of at least one DBM (either an adult or a larva) in a greenhouse. Hence, we based our threshold upon this farming practice, and measured percentage of greenhouses with more than 1 DBM or *C. vestalis*. We have now explained this in two parts of the methods section: lines 118-130 and lines 185-193.

Lines 118-130

“Farmers used prophylactic pesticides at the time of sowing to control the striped flea beetle, *Phyllotreta striolata*. Subsequently, the conditions inside the commercial greenhouses where this study was conducted were controlled by each farmer, based on advice from the local agricultural experiment station, with some control measure implemented upon detection of at least one DBM (either an adult or a larva) in a greenhouse. When farmers found DBM adults or larvae in their greenhouses, they removed them physically (manually or with a trapping device). Only when the densities of DBM or other pest insects were so high as to make physical control impossible did they use pesticides or solarization (i.e., abandoning the crop and using solar radiation to make the internal greenhouse temperature high enough to exterminate pest insects). Unfortunately, interviews with farmers did not provide a detailed history of pest control for each greenhouse except in the case of solarization. We excluded the data from the solarized greenhouses from the analyses.”

Lines 185-193

“As mentioned in 3.2, upon detection of at least one DBM (either adult or larva) in a greenhouse, some control measures (removing them manually or with a trapping device) were likely to be implemented by farmers, although exact details of control measures for each greenhouse were unavailable. Therefore, to compare the treated and untreated greenhouses, we did not use the numbers of DBM and *C. vestalis* in the treated and untreated greenhouses at a particular time in the observation period. Rather, to minimize the effect of pest control by farmers, we compared the relative numbers of greenhouses per month in which more than one DBM adult (monthly incidence

of DBM) or more than one *C. vestalis* (monthly incidence of *C. vestalis*) were found.”

Comment 5.

From a presentation perspective the manuscript needs to be edited both for content and for English. With respect to content, the information on other species observed (start of results) really adds nothing.

Response 5.

We felt it was relevant to explain that DBMs are not the only pest species present in the tested greenhouses. However, in accordance with your feedback, we have moved this information into the Materials and Methods section:

Line 174-181 (Materials and Methods)

“In the Miyama area, we detected not only DBM but also other pest insects of mizuna crops including several aphid species, striped flea beetles (*P. striolata*: Coleoptera: Chrysomelidae), vegetable weevils (*Listroderes costirostris*: Coleoptera Curculionidae), and cabbage armyworms (*Mamestra brassicae*: Lepidoptera: Noctuidae). Some of these insects were occasionally trapped on sticky trap-sheets; however, as the target insect pest was DBM, other insect species on the sheets were not counted. Hymenopterans other than *C. vestalis* trapped on the sticky trap-sheets were also not counted.”

Comment 6.

Also, Figure 1 seems somewhat unnecessary as the data are also in Figure 2.

Response 6.

We have now deleted our original Figure 1, and instead explain the differences between the treated and untreated greenhouses in the text.

Comment 7.

Throughout rate is used when in fact it is incidence, as there is no time function associated with the data.

Response 7.

We have amended the text, and now use the word 'incidence' throughout.

Appendix B

Dear the editor and reviewers

The followings are the point-by-point responses to the comments by the reviewers.

Reviewer: 1

The authors have done a nice job of addressing my concerns. I recommend that the manuscript be accepted for publication.

Reviewer: 2

From a technical perspective I believe they authors have addressed the reviewers' comments in a satisfactory manner, and feel the information certainly merits publication.

However, the text would benefit from editing for presentation. For example, in the first paragraph of the results lines 255-261 could be "In both 2006 (Table 1A) and 2008 (Table 1B) the incidence of DBM was significantly affected by treatment and month, but there were no significant treatment x month interactions"

Response

We changed the sentences according to the comment (Line 22-226).

We also checked the text throughout and made some changes which are shown in red fonts.

This revised version was then edited by the editing company "Editage".